# Heterogeneous absorption of antimicrobial peptide LL37 in *Escherichia coli* cells enhances population survivability

Mehdi Snoussi[1], John Paul Talledo[2], Nathan-Alexander Del Rosario[1], Salimeh Mohammadi[2], Bae-Yeun Ha[3], Andrej Košmrlj[4,5]*, Sattar Taheri-Araghi[2]*

[1]Department of Biology, California State University, Northridge, United States; [2]Department of Physics, California State University, Northridge, United States; [3]Department of Physics and Astronomy, University of Waterloo, Waterloo, Canada; [4]Department of Mechanical and Aerospace Engineering, Princeton University, Princeton, United States; [5]Princeton Institute for the Science and Technology of Materials, Princeton University, Princeton, United States

**Abstract** Antimicrobial peptides (AMPs) are broad spectrum antibiotics that selectively target bacteria. Here we investigate the activity of human AMP LL37 against *Escherichia coli* by integrating quantitative, population and single-cell level experiments with theoretical modeling. We observe an unexpected, rapid absorption and retention of a large number of LL37 peptides by *E. coli* cells upon the inhibition of their growth, which increases population survivability. This transition occurs more likely in the late stage of cell division cycles. Cultures with high cell density exhibit two distinct subpopulations: a non-growing population that absorb peptides and a growing population that survive owing to the sequestration of the AMPs by others. A mathematical model based on this binary picture reproduces the rather surprising observations, including the increase of the minimum inhibitory concentration with cell density (even in dilute cultures) and the extensive lag in growth introduced by sub-lethal dosages of LL37 peptides.
DOI: https://doi.org/10.7554/eLife.38174.001

*For correspondence:
andrej@princeton.edu (AK);
sattar.taheri@csun.edu (ST)

**Competing interests:** The authors declare that no competing interests exist.

## Introduction

Antimicrobial peptides (AMPs) are natural amino-acid based antibiotics that are part of the first line of defense against invading microbes in multicellular systems (*Zasloff, 2002*; *Brogden, 2005*). In humans, AMPs are found in many organs that are in contact with the outside world, including airways, skin, and the urinary tract (*Hancock and Lehrer, 1998*; *Zasloff, 2002*; *Brogden, 2005*; *Jenssen et al., 2006*; *Ganz, 2003*; *Epand and Vogel, 1999*). The short sequence of the AMPs (typically <50 amino acids) along with the flexibility in the design and synthesis of new peptides has spurred attention towards understanding the detailed mechanism of AMPs action which can lead to the rational design of novel antibiotic agents (*Zasloff, 2002*; *Brogden, 2005*; *Hancock and Sahl, 2006*).

A hallmark of the AMPs antibacterial mechanism is the role of physical interactions. Structures of AMPs exhibit two common motifs: cationic charge and amphiphilic form (*Zasloff, 2002*; *Brogden, 2005*). The cationic charge enables them to attack bacteria, enclosed in negatively charged membranes, rather than mammalian cells, which possess electrically neutral membranes. The amphiphilic structure allows AMPs to penetrate into the lipid membrane structures (*Matsuzaki et al., 1995*; *Shai, 1999*; *Ludtke et al., 1996*; *Heller et al., 2000*; *Taheri-Araghi and Ha, 2007*; *Huang, 2000*; *Yang et al., 2001*).

**eLife digest** Many organisms use molecules called antimicrobial peptides as a first line of defense against harmful bacteria. For example, in humans, these natural antibiotics are found in the airways, on the skin and in the urinary tract. Because antimicrobial peptides are positively charged, they function in a different way compared to conventional antibiotics. They are attracted to the negatively charged surface of a bacterium, where they latch onto and penetrate through the membrane that encapsulates the microbe. While this mechanism is well studied at the molecular level, very little is known about how antimicrobial peptides spread and interact in a population of bacteria.

Snoussi et al. combined several approaches to investigate the dynamics of antimicrobial peptides in *Escherichia coli* populations of varying densities. Experiments on single cells showed that peptides stopped the growth of bacteria, which were found to be more susceptible during the late stages of their life cycle. The dying cells then absorbed and retained a large number of antimicrobial peptides. This left fewer 'free' peptides that could target the other *E. coli* cells. In fact, when there were not enough peptides to kill all the bacteria, two sub-populations quickly emerged: one group that had stopped dividing – 'soaking up' the peptides – and another group that could grow unharmed. This new type of cooperation between threatened *E. coli* bacteria is passive, as it does not rely on any direct interactions between cells. The results by Snoussi et al. are relevant to medicine, because they highlight the relative importance for the body to produce enough new antimicrobial peptides to replenish the molecules trapped in bacteria.

DOI: https://doi.org/10.7554/eLife.38174.002

Despite our detailed knowledge about interactions of AMPs with membranes, we lack a comprehensive picture of the dynamics of AMPs in a population of cells. We are yet to determine the extent to which the physical interactions of AMPs disrupt biological processes in bacteria and the degree to which electrostatic forces govern the diffusion and partitioning of AMPs among various cells. Specifically, it was suggested by Matsuzaki and Castanho *et al.* that the density of cells in a culture can alter the activity of AMPs through distributions among different cells (*Matsuzaki, 1999*; *Melo et al., 2009*). We have recently examined the role of adsorption on various cell membranes theoretically (*Bagheri et al., 2015*). Experimental investigations using bacteria and red blood cells by Stella and Wimley groups (*Savini et al., 2017*; *Starr et al., 2016*) directly demonstrated the decisive role of cell density on the effectivity of antimicrobial peptides.

In this work, we utilize complementary experimental and modeling approaches to understand the population dynamics of activity of AMPs from a single-cell perspective. Like all antibiotic agents, AMPs need a minimum concentration (MIC) to inhibit growth of a bacterial culture. For some antibiotics, including AMPs, the MIC is dependent on the cell density. Often referred to as the 'inoculum effect', these phenomena are a trivial consequence of overpopulated cultures. However, in dilute cultures, MICs have been reported to reach a plateau independent of cell density (*Savini et al., 2017*; *Starr et al., 2016*; *Udekwu et al., 2009*; *Artemova et al., 2015*), unless the cell population becomes so small that stochastic single-cell effects become important (*Coates et al., 2018*).

For a precise measurement of the inoculum effect, we extended microplate assays by *Wiegand et al. (2008)* to obtain a functional form of the MIC in terms of the initial cell density (the 'inoculum size'). Contrary to our expectations, we observed that the MIC for the LL37 peptide (AnaSpec, California) remains dependent on *Escherichia coli* density, even in dilute cultures where the average cell-to-cell distance is above 50 μm, much greater than the average cell dimensions (~ 1 ×1×5 μm) (*Taheri-Araghi et al., 2015*). With no direct interactions among the cells and nutrients in excess for all, this dependence suggests that the *effective* peptide concentration is somehow compromised in a cell density dependent manner.

By tracking a dye-tagged version of LL37 peptide (5-FAM-LC-LL37, AnaSpec), we found that the inhibition of growth of *E. coli* cells was followed by the translocation of a large number of AMPs into the cytoplasmic compartment of cells, thus reducing the peptide concentration in the culture, which works in favor of other cells. In the sense of such dynamics, MIC refers to a sufficient concentration of AMPs for absorption into all the cells. Below the MIC, peptide is absorbed by only a fraction of

cells, leaving an inadequate amount of AMPs to inhibit the growth of remaining cells. We have directly observed that cultures with sub-MIC concentrations of dye-tagged LL37 peptides (5-FAM-LC-LL37) exhibit a heterogenous population combining non-growing cells containing many LL37 peptides and growing cells without LL37 peptides.

## Results

### The MIC increases as a function of cell density

The MIC of an antibiotic may depend on the cell density for various reasons: the distribution of antibiotic molecules among bacteria (*Udekwu et al., 2009*; *Clark et al., 2009*; *Melo et al., 2009*; *Jepson et al., 2016*); or as a result of cellular enzyme secretion (e.g., $\beta$-lactamase in case of lactame resistant antibiotics (*Clark et al., 2009*; *Artemova et al., 2015*); or due to the chemical composition of the culture and regulation of gene expression (for instance in late exponential or stationary phases) (*Karslake et al., 2016*; *Artemova et al., 2015*).

In this work we focus on dilute cultures where the dependence of MIC on inoculum size reflects the number of antimicrobial molecules either absorbed or degraded by each individual cell. The MIC for AMPs is in the micromolar range, $\sim 10^{14}$ AMPs/ml. Early exponential cultures contain $\sim 10^6$ cells/ml, which amounts to the ratio of $\sim 10^8$ AMPs/cell. At such a high ratio, only binding or degradation of AMPs of the same order of magnitude per individual cells can lead to the inoculum effect (*Starr et al., 2016*; *Savini et al., 2018*).

To map out the functional form of the inoculum effect, we implemented a two-dimensional dilution scheme on a 96-well plate with a linear dilution of LL37 peptides in columns 7 and 12 followed by a 2/3 dilution series of cells and LL37 peptides over two distinct regions, columns 12 to 8 and 7 to 1 (*Figure 1A*). (See Materials and methods and Appendix 1 for the details of the cell counting and plate preparation). An early exponential *E. coli* culture in rich defined media (RDM, Teknova) was diluted to specific cell densities to cover a relatively even distribution of inoculum sizes.

Each well on the microplate corresponds to a unique combination of densities of LL37 peptides and *E. coli* cells (*Figure 1B*). The cultures in the wells were monitored for 24 hr by an automated plate reader (EPOCH 2, BioTek) in terms of optical density at 600 nm wavelength ($OD_{600}$), while the plate was incubated with orbital shaking at 37°C. Growth or inhibition of growth in each well is evidently distinguishable: growing cultures reach a yield comparable to each other but the non-growing cultures do not show any consistent increase in $OD_{600}$ (*Figure 1C*).

The results, averaged over four similar trials, demonstrate a distinct increase of the MIC as a function of inoculum size (*Figure 1B*). (Detailed data presented in *Figure 1—figure supplement 1*) The solid data points in *Figure 1B* refer to the wells with a growing culture and different marker symbols refer to the number of repeated trial outcomes resulting in growing cultures. A theoretical model developed later in this work nicely fits the average MIC. A separate set of experiments with dense cultures showed that the MIC increases to $3.69 \pm 0.43$ µM and $7.09 \pm 1.88$ µM for the inoculum sizes of $12.2 \times 10^6$ and $24.4 \times 10^6$ cells/ml (8 replicates with three biological repeats were used for each reported value). This is one order of magnitude increase in the MIC value, which can be critical in medical applications.

### Sub-MIC cultures exhibit delayed growth, not slow growth

An interesting feature observed in the results was the extended lag phase, up to several hours, introduced by the sub-MIC concentrations of LL37 peptides (see *Figure 1C*). Despite such growth delay, the average doubling time of the cells ($T_D$) did not change significantly, remaining under 30 min in most cases (see *Figure 1D*). We tested and confirmed the stability of peptides over the duration of the experiment (Appendix 2). Hence, we hypothesized that this behavior was attributed to heterogenous cell death, where the growth of a *fraction* of the cells is inhibited, while the rest of the cells recover the normal population growth after a time delay that is correlated to the number of dead cells. This hypothesis is investigated further at the single-cell level.

### *E. coli* cells absorb and retain peptides

In the microplate experiments, direct cell-to-cell interactions are minimal as the cells are on average over 50 µm apart from each other (corresponding to inoculum size in *Figure 1A,B*). All the

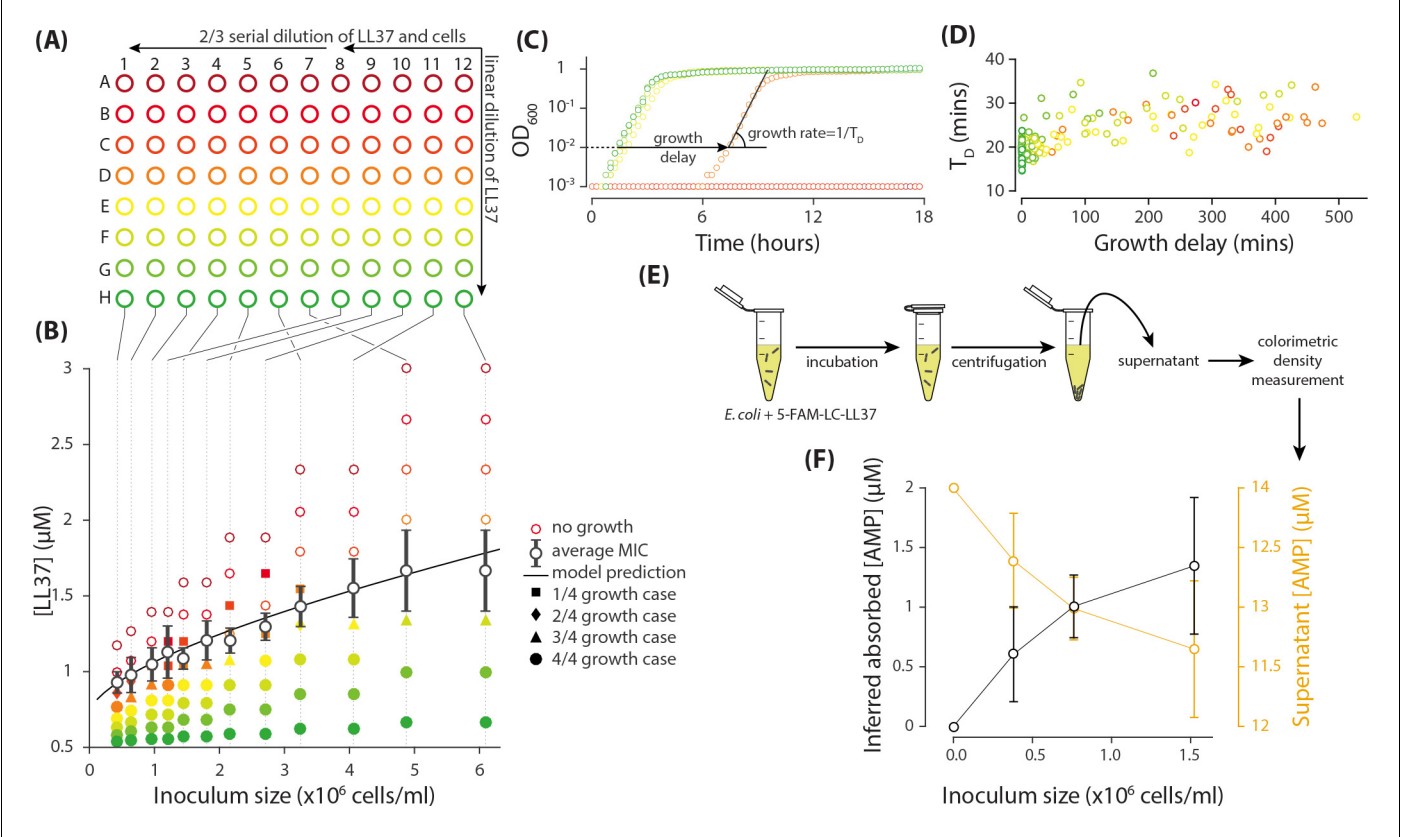

**Figure 1.** Measurement of the inoculum effect and peptide absorption by *E. coli* cells. (A) A two-dimensional dilution scheme, which includes a linear dilution of LL37 peptides in columns 7 and 12 followed by two separate 2/3 dilution series of the cells and LL37 peptides on columns 12 to 8 and 7 to 1. (B) Each well represents a different combination of densities of LL37 peptides and *E. coli* cells from which we can extract the MIC as a function of inoculum size by monitoring growth of the culture in individual wells. The solid data points refer to the wells with growing culture and different marker symbols refer to the number of repeated trial outcomes that resulted in growing cultures. The empty data points refer to wells with no visible growth. A theoretical model developed later in this work nicely fits the average MIC. Data represent four biological repeats where the average and standard deviations of MIC are depicted with black symbols and lines. (C) The growth of the cultures were monitored by an automated plate reader in terms of $OD_{600}$. Growing cultures reach a yield comparable to each other while non-growing cultures do not exhibit consistent increase in $OD_{600}$. Data are examples from column 11 of *Figure 1AB* and they follow the same color coding. (D) Analysis of the growth in sub-MIC cultures reveal that growth is delayed depending on the LL37 concentration, but the doubling time of the cells shows no considerable change. Data are from the same experiments as in panel B and follow the same color coding as panels A and B. (E) Through colorimetric measurement of the concentration of a fluorescently tagged analogue of LL37 peptide (5-FAM-LC-LL37), we can quantify the amount of peptides remaining in the supernatant after incubation with *E. coli* cells. (F) The amount of 5-FAM-LC-LL37 peptides remaining in the supernatant decreases with inoculum size (the initial AMP concentration is 14 µM). The amount of absorbed AMPs by the cells are inferred by subtracting the final (supernatant) from the initial concentration of AMPs. The results are the average of 4 biological repeats. Average and standard deviations are depicted in the figure.

DOI: https://doi.org/10.7554/eLife.38174.003

The following figure supplements are available for figure 1:

**Figure supplement 1.** The full experimental data obtained from the microplate reader consisting of the four trials performed as reported in *Figure 1* of the main text.

DOI: https://doi.org/10.7554/eLife.38174.004

**Figure supplement 2.**

DOI: https://doi.org/10.7554/eLife.38174.005

electrostatic interactions are completely shielded. We asked whether the inoculum effect is due to the absorption of peptides into the cells (*Clark et al., 2009*). AMP absorption into bacteria has been previously discussed and quantified using various techniques. Different prokaryotes were reported to absorb $1 - 20 \times 10^7$ AMPs/cell (*Steiner et al., 1988*; *Savini et al., 2018*; *Starr et al., 2016*; *Tran et al., 2002*; *Melo et al., 2011*; *Roversi et al., 2014*), which is high enough to initiate the inoculum effect. Here we also quantified the absorption of a dye-tagged analogue of LL37 (5-FAM-LC-

LL37, AnaSpec) by colorimetric measurement of 14 µM of AMPs before and after incubation with *E. coli* and separation by centrifugation (see *Figure 1E,F* and Appendix 3 for details). We observed a reduction in the supernatant concentration of AMPs that was proportional to the inoculum size with an average rate of $7.6 \pm 2.1 \times 10^8$ AMPs/cell (calculated based on a linear fit to the data). Since the colorimetric measurements rely on clear solutions, we had to infer the amount of absorbed AMPs by subtracting the supernatant concentration from the initial concentration (*Figure 1F* left axis). Note that the amount of absorbed AMPs in cultures with AMP concentrations higher than the MIC can well exceed the *required* amount of AMPs to kill a cell.

## Single-cell data demonstrate absorption and retention of peptides in target cells

We further investigated peptide absorption by tracking dye-tagged peptide action on live cells. To this end, we brought *E. coli* cells from an exponential culture to an imaging platform where they were treated with an above-MIC concentration (10 µM) of 5-FAM-LC-LL37 under agarose gel containing RDM growth media.

We closely monitored cell growth and distribution/localization of peptides by phase contrast and fluorescent time-lapse microscopy (*Figure 2A*). By analyzing 383 cells, we observed that the inhibition of growth is followed by a rapid translocation of peptides into target cells, as quantified by a jump in the cell's fluorescent signal (*Figure 2B*). As a result, fluorescent signals showed a bimodal distribution over the course of the experiment. A large degree of temporal, cell-to-cell heterogeneity was also observed as the peptide translocation time varied for about 30 min for different cells (*Figure 2B*). The fluorescent signal remained high after translocation confirming retention of peptides in the cells. The density of bound AMPs, however, cannot be estimated from the fluorescent signal due to the possibility of self-quenching of the dyes. As such, the signal strength is reduced in the regions with densely packed fluorophores (*Swiecicki et al., 2016*).

The instantaneous growth rate of individual cells was non-monotonic and data collapsed onto each other once plotted with reference to peptide translocation time (*Figure 2C*). There is a drop to negative values, indicating the shrinking of cells, which was found to be synchronized with the uptake of peptides. The growth rate reaches a steady value of zero in ~10 min.

As a whole, *E. coli* cells exhibited a binary physiological state over the course of the peptide action in terms of growth rate and peptide uptake. That is, the cells were found to be in either of these distinct states: (1) growing, with no significant peptide uptake; and (2) non-growing, followed by an abrupt peptide uptake. This is quantitatively evident in the scatter plot of the instantaneous growth rate as a function of fluorescence intensity, where cells segregate into two separate clusters (*Figure 2D*). At the peptide concentration of 10 µM (above the MIC) all cells were initially in state (1) and then transitioned to state (2) within one generation.

## Growth inhibition is heterogeneous in sub-MIC cultures

Population-level data from microplates were suggestive, but not conclusive, of a heterogeneous growth inhibition in cultures with a sub-MIC concentration of AMPs. Hence, we proceeded with single-cell experiments as noted above with a sole modification of using a lower, sub-MIC concentration of 5-FAM-LC-LL37 (4.0 µM for the dye-tagged peptide). As such, most individual cells grew to form micro-colonies.

The striking observation was the phenotypic heterogeneity in the isogenic population of cells in each micro-colony (*Figure 3A*). As a colony expanded, growth of some cells was inhibited and peptides translocated into them. We observed a similar transition from state (1) to state (2) as previously seen in above-MIC cultures (*Figure 2*). The difference is that at sub-MIC the transition occurred for only a fraction of the cells.

Analysis of 13 separate micro-colonies, consisting of a total of 280 cells (over the course of the experiment), showed that the relative size of the colonies initially increased exponentially until the appearance of non-growing cells (*Figure 3B*). The fluorescence intensity of non-growing cells showed an abrupt transition, as in the case of above MIC cultures, with comparable relative changes in the signal, which suggests that a similar number of peptides are taken by each cell (*Figure 3C*).

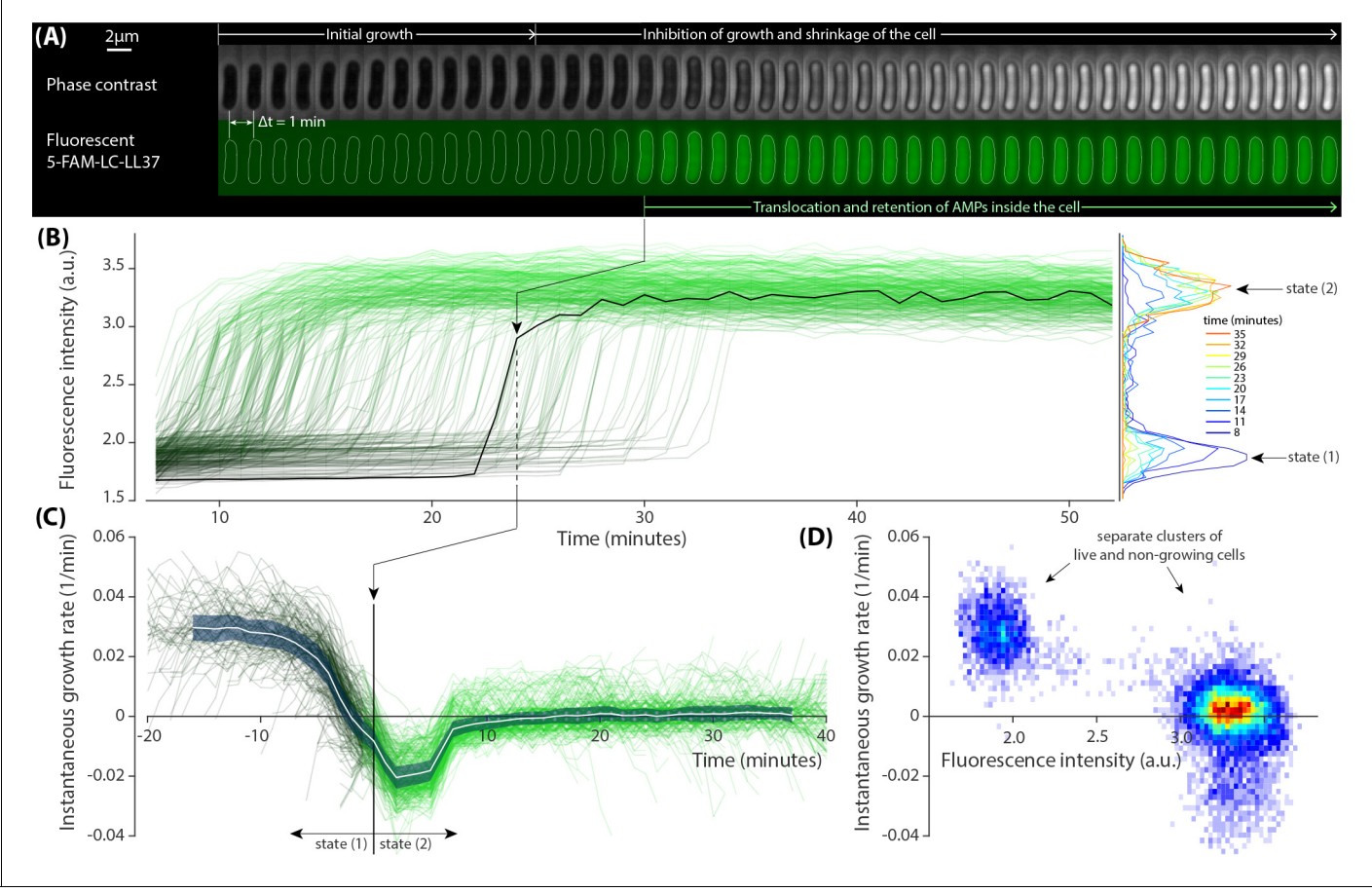

**Figure 2.** Growth inhibition of *E. coli* cells by dye-tagged LL37 peptides. (**A**) Sample phase contrast and fluorescent time-lapse images of an *E. coli* cell treated with a lethal dosage of the dye-tagged LL37 peptides (5-FAM-LC-LL37). Phase contrast images show inhibition of growth and slight shrinkage of the cell after a brief growth period. The fluorescent channel shows the distribution, translocation and retention of the peptides. (**B**) Abrupt transition in the fluorescent signal of 383 cells that occur over a span of more than 30 min. The black line corresponds to the sample shown in panel A. The fluorescence intensity histogram is depicted for different time points on the right. (**C**) The instantaneous growth rate of individual cells collapse on each other when plotted in referenced to the peptide translocation point. The average behavior (white line) of the collapse shows that the translocation happens shortly after the inhibition of growth and shrinkage of the cell. The shaded area denotes the standard deviation. (**D**) The two dimensional distribution of instantaneous growth rate and fluorescent signal from all time points depict well-separated clusters referring to a binary response of the cells to the peptides.

DOI: https://doi.org/10.7554/eLife.38174.006

## Mathematical model based on peptide absorption reproduces and explains experimental observations

In order to test whether the absorption of peptides can explain the inoculum effect, we developed a mathematical model with minimal single-cell assumptions (*Figure 4A*), which describes the time evolution of the mean concentration $[B]$ of growing bacteria and the mean concentration $[P]$ of free AMPs in the solution. The model describes two processes: (1) bacteria are assumed to divide with a constant rate $k_D = \ln 2 / T_D$, where $T_D \approx 23\,min$ is the average doubling time; (2) AMPs kill growing bacteria with a rate $k_k$, and afterwards each dead cell quickly takes up $N$ AMPs (see *Figures 2B* and *3C*). These AMPs are bound to the membrane as well as to the cytoplasm of the cell and are not recycled to attack other cells. The time evolution of concentrations of growing bacteria $[B]$ and available AMPs $[P]$ is described by the following equations:

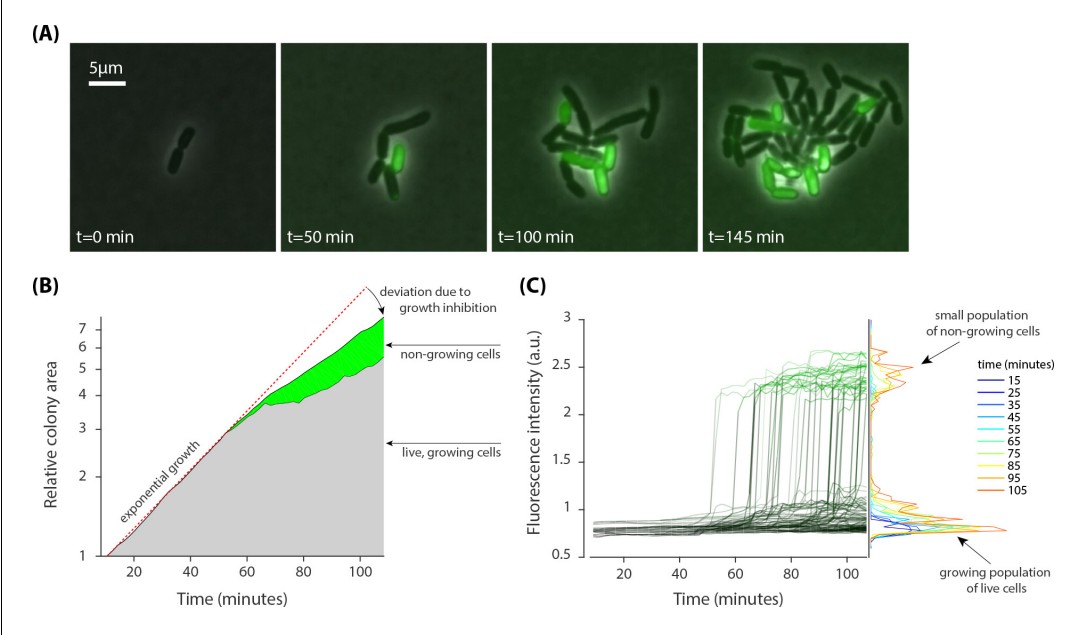

**Figure 3.** Heterogenous growth inhibition and peptide absorption at a sub-MIC concentration of peptides. (**A**) Sample time-lapse images (overlay of phase contrast and fluorescent image) of dividing *E. coli* cells show that growth of only some cells is inhibited in a growing colony. Phase contrast images show growth of a micro-colony and the fluorescent channel (green) shows translocation of the dye-tagged LL37 peptides (5-FAM-LC-LL37) in the cells whose growth is inhibited. (**B**) Relative growth of the total area of 13 separate colonies, consisting of 280 cells, depicts initial exponential expansion (grey area) until the appearance of non-growing subpopulation (green area). (**C**) An abrupt transition in fluorescent signal is observed when growth is inhibited in cells. The transition for individual cells is similar to that in above MIC cultures (see *Figure 2*).

DOI: https://doi.org/10.7554/eLife.38174.007

$$\frac{d[B]}{dt} = k_D[B] - k_k[B][P], \qquad (1)$$

$$\frac{d[P]}{dt} = -Nk_k[B][P]. \qquad (2)$$

This model predicts two different outcomes (see *Figure 4B*) depending on the initial concentrations of bacteria and AMPs: (1) The population of bacteria goes extinct for a sufficiently large concentration of AMPs, that is above MIC; (2) The population of bacteria can recover in a low concentration of AMPs, that is below MIC. The two unknown parameters of the model were fitted to best approximate the MIC dependence on the inoculum size (*Figure 4C*). The fit resulted in a killing rate $k_k = 0.040\,\mu M^{-1}\,min^{-1}$ and $N = 3.8 \times 10^7$ AMPs absorbed per dead cell. Note that the estimated number of absorbed AMPs from the fitting is much smaller than the estimated number of $7.6 \pm 2.1 \times 10^8$ AMPs/cell from the colorimetric measurements. This discrepancy might arise from the fact that cells can divide several times before they stop growing and absorb peptides, which was neglected in the estimate from the colorimetric measurements.

Note that in the limit, where the initial concentration of bacteria goes to zero, the MIC value approaches the finite value $k_D/k_k = 0.75\,\mu M$ (see *Equation 1*). This is consistent with a previous model by Stella group (*Savini et al., 2017*), which considered that bacteria get killed once the number of peptides bound to cell membrane reaches a certain threshold. This threshold is expected to be smaller than the number $N$ of absorbed AMPs by dead cells, which includes peptides bound to cell membrane as well as peptides bound to intracellular content. (Note that the number of surface bound peptides correlates with the concentration of free peptides in solution).

We further examined whether the model can reproduce other experimental data without additional fitting. In particular we tested whether the model could predict the growth delay in surviving bacterial population when the concentration of AMPs is increased (see *Figure 1C*). The predictions of our model agree reasonably well with experimental results for the growth delay of population

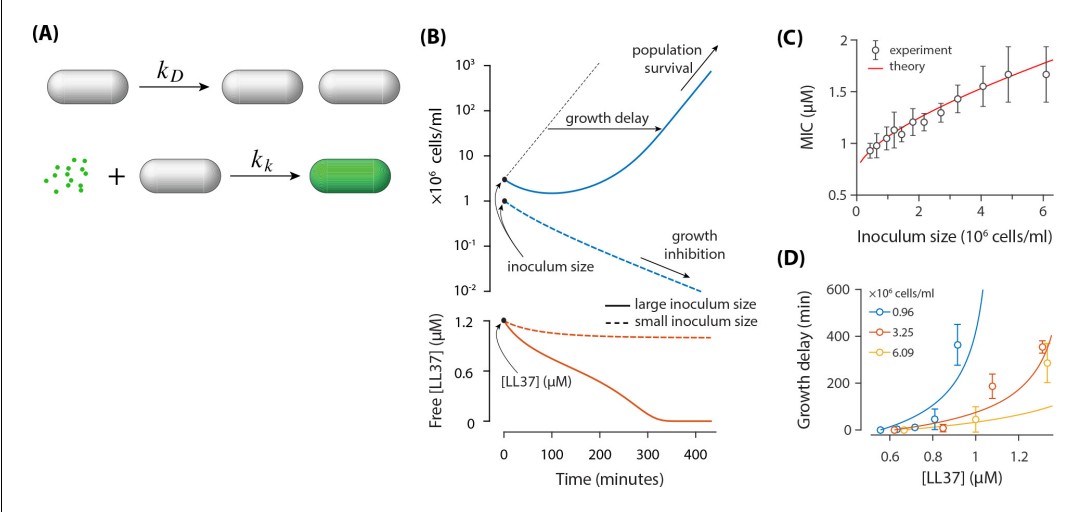

**Figure 4.** A theoretical model based on the absorption of peptides in *E. coil* cells. (**A**) *E. coli* cells replicate with a rate of $k_D$ and they get killed with a rate of $k_k$ Each dead cell quickly absorbs *N* AMPs. (**B**) Demonstration of the inoculum effect for the initial AMP concentration [LL37]=1.2 µM. A culture with high inoculum size ($3 \times 10^6$ cells/ml) survives as peptides deplete (solid lines) whereas growth in small inoculum size ($10^6$ cells/ml) is inhibited with excess peptides remaining in the solution (dashed lines). Despite the survival of culture with high inoculum size, the growth is delayed. (**C**) Comparison of the MIC between the theoretical model and the experimental data. (**D**) Growth delay as a function of [LL37]. Solid lines are theoretical results (not a fit) and circles represent experimental data from microplate experiments (*Figure 1*). The delay is calculated with respect to the lowest AMP concentration (row H) of *Figure 1AB*.

DOI: https://doi.org/10.7554/eLife.38174.008

(see *Figure 4D*), given the simplicity of the model. Deviations likely originate from the fact that cell doubling times are scattered (*Figure 1D*), while our model assumes a fixed cell doubling time of 23 min. Due to the exponential growth, the growth delay is highly sensitive to the cell doubling time.

## The action of dye-tagged LL37 peptides is cell-cycle and cell size dependent

The temporal heterogeneity we reported in growth inhibition and peptide retention is a key factor for the emergence of the surviving subpopulation. The wide distribution of ~ 30 min (above MIC cultures, *Figure 2B*) is puzzling, as all the cells experienced the same environmental conditions. We looked at the correlations of the translocation time with two cell size measures to investigate any dependence on the physiological conditions of the target cells. A negative correlation was observed between peptide translocation time and initial cell length, indicating that small cells can resist the action of AMPs and they continue growing until a later time point (*Figure 5A* left panel). In contrast, cell length at translocation time did not show any correlation with the translocation time (*Figure 5A* right panel). This clearly demonstrated that the action of 5-FAM-LC-LL37 is cell-cycle and cell-age dependent. This seems in agreement with the findings made by the Weisshaar lab, where LL37 peptides were observed to first bind to the septum of dividing cells. Thus, the higher chance to act on larger, dividing cells, as opposed to small growing cells (*Sochacki et al., 2011*). The stronger binding of LL37 peptides to the septum area is not well understood but may have multiple physical origins. Among various possibilities, the Wong lab has shown that LL37 peptides preferentially bind to membranes with negative Gaussian curvature, a geometry that can be found in the septum of rod shaped microorganisms (*Yang et al., 2008*; *Schmidt et al., 2011*).

To investigate the cell-size dependence of action of AMPs we added a sublethal dosage of cephalexin to the growth media (RDM) that was shown to slightly increase cell sizes without the loss of viability by delaying cell divisions (*Si et al., 2017*). Our measurements using the microfluidic 'mother machine' (*Wang et al., 2010*) also showed that 1 µg/mL of cephalexin results in the 17% increase in the length of the newborn cells (*Figure 5B*), from 3.37 µm to 3.96 µm, without any appreciable change of the growth rate. Based on the microplate experiments, the MIC of cells grown in the

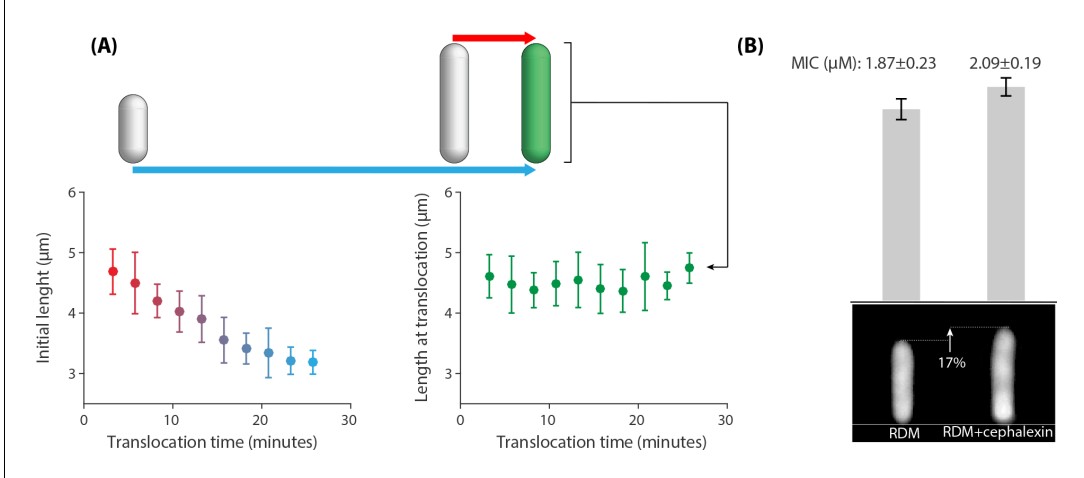

**Figure 5.** Correlations between the peptide translocation time and cell size for 383 cells that were treated with an above-MIC concentration of 5-FAM-LC-LL37 peptides reported in *Figure 2*. (A Left panel) A negative correlation between initial cell length and AMP translocation time indicates resistance of small cells to peptides until a later time points in their life cycle. (A Right panel) Cell length at translocation time and the time of translocation are not correlated. Average and standard deviation are depicted for the data binned based on the x axis. (B) Average cell size affects the MIC values. Addition of 1µM/mL of cephalexin to the growth media (RDM) increases cell size by 17% (sample newborn cells shown in the bottom panel), which resulted in an increase in the MIC. The inoculum size of $6.1 \times 10^6$ cells/ml are used for this experiment. Twelve replicates with three biological repeats were used for this data.

DOI: https://doi.org/10.7554/eLife.38174.009

presence of cephalexin increased from $1.87 \pm 0.23$ µM to $2.09 \pm 0.19$ µM for those grown in regular RDM at the inoculum size of $6.1 \times 10^6$.

## Discussion

Heterogeneities in bacterial response to antibiotics can be critical if leading to the survival of a sub-population that can recover population growth (*Coates et al., 2018*). In this work, we discovered an unexpected absorption and retention of an antimicrobial peptide (LL37 and the dye-tagged ana-logue, 5-FAM-LC-LL37) in *E. coli* cells, which under sub-MIC concentrations led to the emergence of two distinct subpopulations in an isogenic bacterial culture: a group of cells that retain peptides after their growth is inhibited and a group of surviving cells that grow owing to the reduction of the 'free' peptide concentration by the other group. This 'passive cooperation' is an interesting feature of an *E. coli* culture where cells do not have any form of active communication, unlike ion-channel based cooperation in *Bacillus subtilis* biofilms (*Prindle et al., 2015*; *Liu et al., 2017*; *Liu et al., 2015*).

At the cellular and molecular scales, a distinct feature of the action of AMPs is its collective nature, where a large number of AMPs are required to first bind to and then disrupt the cell mem-branes to kill the cell. Absorption of AMPs has been discussed and quantified previously for different microorganisms and peptides using various techniques (*Steiner et al., 1988*; *Savini et al., 2018*; *Starr et al., 2016*; *Tran et al., 2002*; *Melo et al., 2011*; *Roversi et al., 2014*). Utilizing live, single-cell microscopy, we observed the temporal and cell-to-cell heterogeneity in the peptide absorption into *E. coli* cells, which goes beyond the membrane binding.

The integration of the population and single-cell data, combined with the theoretical modeling, presented in this work, provides strong evidence that the inoculum effect in the case of LL37 pepides arises from the retention of these peptides in target cells. Despite the seeming complexity of the partitioning of LL37 peptides in a population of bacterial cells, the picture at the single-cell level is simple and binary, consistent in cultures with above-MIC and sub-MIC concentrations of AMPs: occurrence of growth inhibition depends on the free peptide concentration and is followed by an abrupt, permanent translocation of peptides into the target cells.

Our quantification of peptide retention ($\sim 3.8 \times 10^7$ peptides/cell) is within the range reported using various methods (*Steiner et al., 1988*; *Savini et al., 2018*; *Starr et al., 2016*; *Tran et al., 2002*; *Melo et al., 2011*; *Roversi et al., 2014*). Yet, this large number raises the question of which molecules inside of the cells are interacting with LL37 peptides. The negative charge of DNA as well as some proteins can provide binding sites for LL37 peptides. To categorically distinguish between these two, we utilized an *E. coli* strain lacking the septum positioning *minCDE* system, which produces enucleated mini-cells (*Figure 6A*). The mini-cells do not contain DNA as confirmed by the localization of a fluorescently tagged histone-like protein *hupA* (*Figure 6A*, see Materials and methods for the details of the strains genotype). Translocation of 5-FAM-LC-LL37 peptides into mini-cells showed qualitatively similar absorption and retention as seen in regular cells (*Figure 6B*), which suggests the presence of significant interactions of AMPs with the intracellular content other than DNA. Yet, the rate at which AMPs are absorbed into the mini-cells is slower than that in the neighboring mother cells with DNA content (*Figure 6B*), perhaps indicating the role of the negative charge of DNA.

While our results provide a quantitative picture of LL37 partitioning and acting in an *E. coli* population, they open new questions on the molecular and evolutionary basis of their activity. First, the strong absorption of peptides into the cytoplasmic area raises questions on the nature and impact of this intercellular binding: what specific proteins and domains are peptides binding to? How does the peptide binding perturb the protein functions? Second, the population survivability as a result of peptide absorption raises questions on the evolutionary dynamics of this phenomena: how do *E. coli* cells evolve to achieve this cooperative fit? How does this phenomena affect multi-species cultures with prokaryotic and eukaryotic organisms?

Finally, our findings imply an important dynamics for the activity of LL37 peptides (and possibly other AMPs) in the multicellular host organisms. Considering the peptide absorption into the target cells, AMP concentration should not be assumed the only key factor, but the rate of the expression of the AMPs by the host is also decisive in determining the effectivity of AMPs. The expression rate competes with the rate of absorption of the AMPs in the bacterial cells. In the results presented in this work, we focused on closed systems where the total number of AMPs remained constant. As a future direction, one could examine growth inhibition of bacterial cultures experiencing an influx of the AMPs.

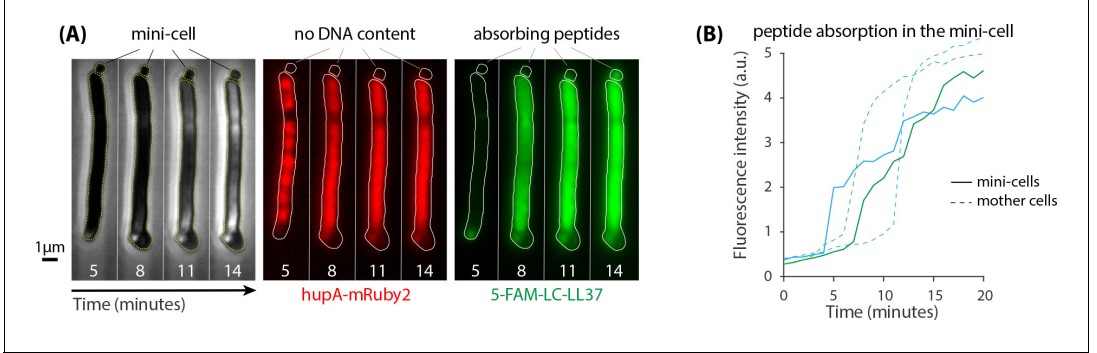

**Figure 6.** Δ*minCDE* strain of *E.coli* was used to test the significance of interactions between peptides and DNA as well as other intracellular content. (A) Lack of septum positioning system produces enucleated mini-cells (left panel) that do not contain DNA as indicated by the localization of *hupA-mRuby2* protein (middle panel). Mini-cells absorb a large amount of 5-FAM-LC-LL37 peptides similar to wild-type cells (right panel) (B) Fluorescence intensity per area of two mini-cells suggests a significant peptide interactions with enucleated mini-cells. The rate of absorption of AMPs in mini-cells is slower than in the neighboring mother cells with DNA content.
DOI: https://doi.org/10.7554/eLife.38174.010

## Materials and methods

### Bacterial strains and growth conditions

In this work, we used derivatives of a prototrophic *Escherichia coli* K12 strain, NCM3722, that was constructed, sequenced, and extensively tested by Kustu and Jun labs (*Soupene et al., 2003*; *Brown and Jun, 2015*). In all microplate and single-cell experiments, we used ST08, a nonmotile derivative of NCM3722 (ΔmotA) (a gift from Suckjoon Jun's Lab at the University of California, San Diego), except for experiments with mini-cell producing strains where we used ST20. This strain possesses a deficiency in septum positioning system (ΔminCDE) and a DNA marker (*hupA-mRuby2*). ST20 was constructed using standard P1 transduction to transfer the gene deletion minCDE::aph from PAL40 (from Petra Levin's Lab at the Washington University, St. Louise) to ST12, a construct from ST08 with the infusion of *mRuby2* fluorescent protein with the histone like protein *hupA* (from Suckjoon Jun's Lab).

In all experiments, a MOPS based rich defined media (RDM) was used, developed by Fred Neidhardt (*Neidhardt et al., 1974*), which is commercially available from Teknova Inc. The average generation time of the *E. coli* strain used in this study was 23 min in RDM at 37°C. All cells in the experimental samples were grown to early or mid log phase prior to the start of the experiment in a 37°C water bath shaker, set to 240 rpm.

#### Sample preparations

The cell culture for each experiment was carried out in three stages: seeding of culture, pre-culture, and experimental culture.

- The seed culture was inoculated from an isolated colony on a Lysogeny Broth (LB) agar plate into 3 ml of RDM in a culture tube (950 mm ×150 mm) containing a loose cap set shaking at 240 rpm in a 37°C water bath to ensure aeration. The colonies on the LB agar plate were prepared by streaking a −80°C glycerol stock on the plate, first incubated at 37°C for 24 hr and then kept in the 4°C fridge for a total of one week.
- For the pre-culture, the seed culture was diluted 1000-fold in the identical growth medium after overnight growth and incubated in the 37°C water bath shaker until it reached mid exponential phase.
- For the experimental culture, the pre-culture at mid-exponential phase was diluted 200-fold in the same growth medium to ensure the cells remained in steady-state growth conditions for approximately 5 hr. After about eight generations, the experimental culture reached mid exponential phase. This culture containing cells in steady-state exponential phase was then diluted to appropriate cell densities for microplate and single-cell experiments.

### Antimicrobial peptides

Antimicrobial peptides used in this study included LL37 and a dye-tagged version 5-FAM-LC-LL37. All peptides were purchased from AnaSpec, California. The net peptide content of the purchased material was quantified at 75% by elemental analysis of the carbon, hydrogen, and nitrogen content (CHN analysis) through the manufacturing company.

The product was shipped in vials as a dry powder. 400 $\mu$M stocks suspended in autoclaved double distilled $H_2O$ were prepared in-house. The stock was stored in a −20°C cold storage until ready for use.

Special handling precautions for the antimicrobial peptides, the stock solution, or solutions containing them were used. Low protein binding supplies including pipette tips (Low Retention Aerosol Barrier, FisherBrand), Eppendorf tubes (Protein LoBind, Eppendurf), and microplates (Ultra-Low attachment, Corning) were used.

### Analysis of microplate data

To analyze the microplate data we developed a custom MATLAB (MathWorks Inc.) code to measure the growth rate and $T_{0.1}$ for each individual well. The latter quantifies the time at which the $OD_{600}$ reaches 0.1, as a parameter that shows a 'delay' in the growth of the culture. The growth rate is calculated in the standard manner by fitting an exponential line against the growth curve that resembles stable, exponential growth.

We have conducted four experimental repeats to construct the data reported in *Figure 1* of the manuscript. The full experimental data for each of these four repeats are depicted *Figure 1—figure supplement 1*, which shows the growth curves for all 96 wells and the calculated values for the growth curve and $T_{0.1}$. Each box in the plot refers to one well, where the vertical axis corresponding to each well denotes the log scale of the $OD_{600}$ (after subtracting the blank value) and horizontal axis resembles the time during the 24 hr the experiment took place. The red curves correspond to the growth curve while the black overlap shows the detected exponential growth region and the blue line represents the exponential fit with respect to the data. The two numbers in each box represent the calculated values for the doubling time (in the top) and the $T_{0.1}$ (in the bottom), both in terms of minutes. The $T_{0.1}$ is calculated from the initial readings generated by the plate reader. The boxes with no number and gray curve refer to wells that did not show any growth.

## Cell counting experiments and calibration

In order to determine the relationship between the MIC and cell density, we first had to find the correlation between cell density and optical density measured at a wavelength of 600 nm ($OD_{600}$). A high precision cell counting experiment was performed using a hemocytometer (Hausser Scientific 3900) and high magnification microscopy. Using different cell cultures with different values of $OD_{600}$, we were able to correlate values obtained for $OD_{600}$ measured on a spectrophotometer (Genesys 20, Fisher Scientific) with the density of *E. coli* ST08 cells.

For the cell counting protocol, cells from the experimental culture at mid exponential phase were harvested, fixed with 0.025% formaldehyde, and diluted to four varying concentrations, corresponding to $OD_{600} = 0.115, 0.066, 0.033,$ and $0.022$. Each of these cultures were transferred to a hemocytometer, allowing for the enumeration of cells. The number of cells where counted in the specified grids of the hemocytometer with dimensions of $50 \times 50 \times 20 \, \mu m^3$. The imaging was performed on a Nikon-Ti microscope with a 40× objective lens using transmission light phase contrast microscopy. For each specific $OD_{600}$ value, we have used 10 fields of view, each containing 42 grids. The histogram of the cell population in each grid is shown in *Figure 1—figure supplement 2A*.

The average number of cells per grid compared with the average density of cells for each culture produced a strong linear correlation with the measured $OD_{600}$ values obtained. The linear regression of the cell density as a function of $OD_{600}$ results in the conversion factor of $6.09 \times 10^8$ cells/ ml (*Figure 1—figure supplement 2B*).

## Live-cell imaging platform

To monitor the inhibition of growth for *E. coli* cells by antimicrobial peptides in a live microscopy setting we chose to use agarose pads, using 3% low melt agarose gel, to immobilize them. Inspired by previous works (*Moffitt et al., 2012*; *Priest et al., 2017*), we developed a system for patterning and housing specific amounts of agarose gel suitable for long term microscopy needs. The patterns are parallel channels that allow cells to spread and move away from one another under the agarose pad, while aligned in certain directions to help with the image analysis and cell segmentation. The housing also reduces the chance of evaporation of the liquid culture, thus allowing the gel to be used for hours at 37°C during microscopy.

## Phase contrast and fluorescent imaging and analysis

An inverted microscope (Nikon Ti-E) equipped with the Perfect Focus System (PFS 3), a 100x oil immersion objective lens (NA 1.45), and an Andor Zyla sCMOS camera were used for imaging. The light source that was used for the phase contrast microscopy was made possible with the help of LED transmission light (TLED, Sutter Instruments 400–700 nm) and Spectra X light engine (Lumencor), which was used for fluorescent imaging.

The illumination condition for phase contrast was 50 ms exposure with an illumination intensity set to 10% of the max TLED intensity. The fluorescent images for the 5-FAM dye were taken with the excitation wavelength of 485 while using a quad band filter (DAPI/FITC/TRITC/Cy5, 84000, Chroma Technologies).

For temperature control, the microscope was incubated in a custom made plexiglass housing with a forced air heater and temperature controller (Air-Therm SMT, World Precision Instruments) set to $37 \pm 0.1$°C. The temperature was constantly monitored during the course of the experiment.

We developed custom high-throughput image analysis software optimized for segmenting and analysis of individual cells whose growth are inhibited by LL37 or 5-FAM-LC-LL37 under patterned gel agarose gel (*Taheri-Araghi, 2018*; copy archived at https://github.com/elifesciences-publications/CellSegmentation). For the heterogeneous colonies of growing and non-growing cells, the boundary of the cells were first highlighted manually to increase contrast for segmentation.

The procedure includes standard, predefined image enhancement and processing steps available as functions in MATLAB (MathWorks Inc.) image processing toolbox. The overall steps are as follows:

- Step 1: Cropping individual cells and stitching the corresponding time-lapse frames to make a kymograph-like image construction for the each cell.
- Step 2: Using Sobel edge finding function to determine the boundary of each cell both in phase contrast and fluorescent images.
- Step 3: Size based filtering of mis-segmented cells.
- Step 4: Creating a binary image based on the cell Sobel edge data.
- Step 5: Measuring dimensions (width and length) of the cells based on the data from phase contrast image.
- Step 6: Measuring the average fluorescence intensity over the cell segment cells.

The instantaneous growth rate is calculated assuming exponential growth for the cells and taking the derivative of the cell length as a function of time. To decrease the noise in the calculations, an exponential curve was fitted to three consecutive datapoint of cell length as a function of time.

## Analysis of the theoretical model with Mathematica

Differential equations (*Equations 1–2*) describing the population dynamics model were analyzed in Mathematica with the function *NDSolve*. For each initial concentration of bacteria and peptides, the evolution of both populations were analyzed during the subsequent $T_{max} = 1000$ min. If the concentration of peptides $[P(T_{max})]$ at the end of simulation was smaller than $k_D/k_k$, then the bacterial population survived (see *Equations 1–2*). On the other hand, if the concentration of peptides $[P(T_{max})]>k_D/k_k$, then the bacterial population most likely went extinct. The MIC concentration for a given initial concentration of bacteria was obtained by requiring $[P(T_{max})] = k_D/k_k$, which was found using the bisection method. Finally, the unknown model parameters $k_k$ and $N$ were obtained by minimizing the error between the values of MIC from the model and the experimental data (see *Figure 4C*) by using the function *FindMinimum*.

## Acknowledgements

We are grateful to Lorenzo Stella, Gerard Wong, Dana Harmon, Cristian Ruiz-Rueda and Steven Brown for stimulating discussions and critical reading of the manuscript. We thank Siddhartha Sarkar for the help with MATLAB. We acknowledge funding support from the National Institutes of Health grant 1R15GM124640, the Pilot Project grant under 1RL5GM118975, and the National Science Foundation (NSF) grants Research Experiences for Undergraduates (REU) site grant (EEC-1559973), Partnership in Research and Education in Materials (PREM) between the W M Keck Computational Materials Theory Center (CMTC) at California State University, Northridge, and Princeton Center for Complex Materials (PCCM) (DMR-1205734), and the Materials Research Science and Engineering Center Program through the PCCM (DMR-1420541).

## Additional information

### Funding

| Funder | Grant reference number | Author |
| --- | --- | --- |
| National Science Foundation | DMR-1205734 | John Paul Talledo<br>Andrej Košmrlj |
| National Science Foundation | DMR-1420541 | John Paul Talledo<br>Andrej Košmrlj |

| National Science Foundation | EEC-1559973 | Andrej Košmrlj |
| | | John Paul Talledo |
| National Institute of General Medical Sciences | 1R15GM124640 | Sattar Taheri-Araghi |
| National Institute of General Medical Sciences | 1RL5GM118975 | Sattar Taheri-Araghi |

The funders had no role in study design, data collection and interpretation, or the decision to submit the work for publication.

### Author contributions
Mehdi Snoussi, Investigation, Formal analysis, Data curation, Methodology, Writing—review and editing; John Paul Talledo, Investigation, Formal analysis, Data curation, Methodology; Nathan-Alexander Del Rosario, Investigation, Methodology; Salimeh Mohammadi, Investigation, Formal analysis, Methodology; Bae-Yeun Ha, Conceptualization, Validation, Methodology, Writing—review and editing; Andrej Košmrlj, Sattar Taheri-Araghi, Conceptualization, Resources, Data curation, Software, Formal analysis, Supervision, Funding acquisition, Validation, Investigation, Visualization, Methodology, Writing—original draft, Project administration, Writing—review and editing

### Author ORCIDs
Andrej Košmrlj (iD) http://orcid.org/0000-0001-6137-9200
Sattar Taheri-Araghi (iD) http://orcid.org/0000-0001-8736-1893

### Decision letter and Author response
Decision letter https://doi.org/10.7554/eLife.38174.022
Author response https://doi.org/10.7554/eLife.38174.023

# Additional files

### Supplementary files
• Transparent reporting form
DOI: https://doi.org/10.7554/eLife.38174.011

### Data availability
Data is available on Dryad (https://dx.doi.org/10.5061/dryad.p083411) and the image analysis software is provided on GitHub (https://github.com/staheria/CellSegmentation; copy archived at https://github.com/elifesciences-publications/CellSegmentation).

The following dataset was generated:

| Author(s) | Year | Dataset title | Dataset URL | Database and Identifier |
| --- | --- | --- | --- | --- |
| Snoussi M, Talledo JP, Del Rosario N, Ha B, Košmrlj A, Taheri-Araghi S | 2018 | Data from: Heterogeneous Absorption of Antimicrobial Peptide LL37 in Escherichia coli Cells Enhances Population Survivability | https://dx.doi.org/10.5061/dryad.p08341 | Dryad Digital Repository, 10.5061/dryad.p083411 |

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

# Appendix 1

DOI: https://doi.org/10.7554/eLife.38174.012

## The Two-Dimensional Microplate Experiment

To investigate the dependence of MIC on cell density, we developed a two dimensional dilution scheme on a 96-well plate that covered a wide range of LL37 concentrations and cells densities (*Appendix 1—figure 1A,B*). To this end, a microplate was divided into two sections: rows 1 through 7 and rows 8 through 12. A total of 12 different values of cell densities were used. LL37 was diluted in two dimensions in a series of steps, ensuring proper coverage of the targeted range around the MIC with high precision.

The 96-well plate was first filled with the LL37 peptide using a series of vertical and horizontal dilutions (*Appendix 1—figure 1A–C*). At the end of the series, each well contained 50 $\mu$l of solution with twice the amount of final desired peptide concentration. Then a cell culture was diluted to the desired cell density and transferred to the microplate. Detailed steps of the process and the resulting concentrations are depicted in *Appendix 1—figure 1* and are explained in the figure caption.

The plate was incubated at 37°C in a plate reader (EPOCH 2, Biotek). The growth of the cultures in the wells were monitored by measuring the $OD_{600}$ of each well every 15 min for 24 hr. A lid is used to reduce evaporation while the a vertical gradient of +2°C was applied to reduce the amount of condensation on the lid. The plate was left to shake continuously at an orbital speed of 590 rpm.

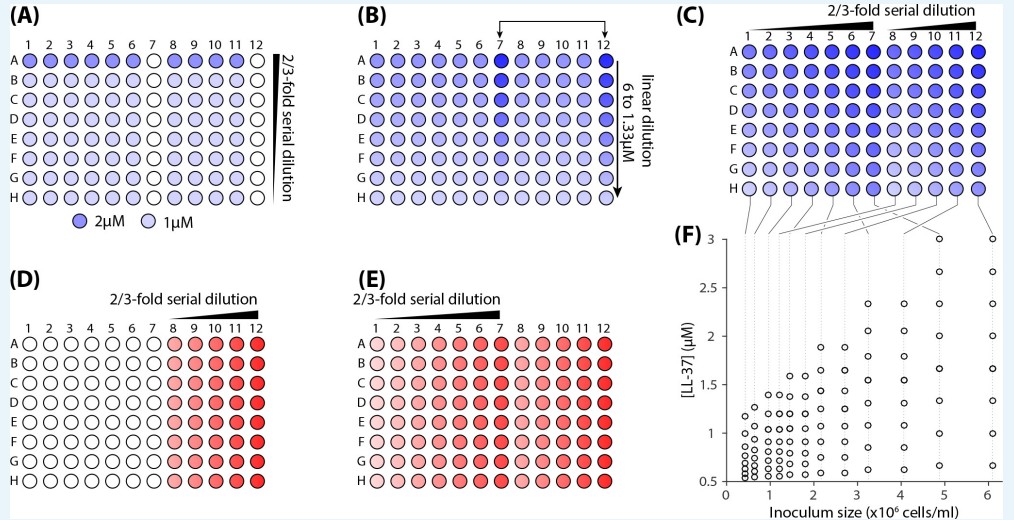

**Appendix 1—figure 1.** Steps for preparing a two-dimensional dilution series of LL37 and cells. (**A**) 50 $\mu$l/well of a 1 $\mu$M solution of LL37 in the growth medium (RDM) was transferred to all of the wells excluding row A, columns 7 and 12. Next, 150 $\mu$l/well of a 2 $\mu$M solution was transferred to the wells of row A, leaving columns 7 and 12 blank. A 2/3-fold serial dilution was then carried out vertically on all rows, A through H. (**B**) 150 $\mu$l/well of a linear dilution series of LL37 was was transferred to columns 7 and 12, with highest concentrations located in row A, while the lowest in row H. The concentrations chosen were 6, 5.34, 4.67, 4, 3.33, 2.67, 2 and 1.33 $\mu$M. (**C**) A 2/3-fold serial dilution was carried out horizontally from Columns 12 through 8 and Columns 7 through 1. The final volume of the solution in each well was 100 $\mu$l. (**D**) A cell culture was diluted to a final $OD_{600}$ of 0.02 and 50 $\mu$l/well was transferred into the wells located in column 12. A 2/3-fold dilution series was performed in a separate reservoir and 50 $\mu$l/well was transferred to column 11. The process was repeated for columns 10, 9, and 8. (**E**) The cell culture diluted to the $OD_{600}$ = 0.016 (80% of the culture used for column 12). 50 $\mu$l/well was transferred into the wells located in column 7. For columns 6 through 1, the

same volume was transferred to each well after repeating 2/3-fold dilutions of this culture. (**F**) The final concentration of LL37 and cell densities in the microplate.

DOI: https://doi.org/10.7554/eLife.38174.013

## Blank calculation

The designed plate scheme did not include a 'blank', a well lacking any cells. Thus, for the calibration of the optical density ($OD_{600}$) we correlated the initial $OD_{600}$ readings of the plate reader with the calculated, known initial cell densities in the culture to extrapolate the value of the blank $OD_{600}$. We have a total of 12 different initial cell densities in the 96 well plates. The relationship between the initial $OD_{600}$ of the plate reader and the cell densities are expected to be linear with the intercept corresponding to the blank $OD_{600}$ of the plate reader.

The blank $OD_{600}$ is calculated separately for each of the four experiments we conducted in order to produce the data reported in *Figure 1* of the main text. The data for extrapolating the blank value is reported in *Appendix 1—figure 2*, when the average of the first three readings (30 min intervals) on the plate reader is correlated with the cell density in each well as calculated based on the preparation steps. Only the wells that did not show any final growth were chosen for this calculation, thus avoiding the effect of growth in the initial stage of the experiment. The linear regression for each panel is shown in red, while the values corresponding to the intercept are noted in red beside the arrow.

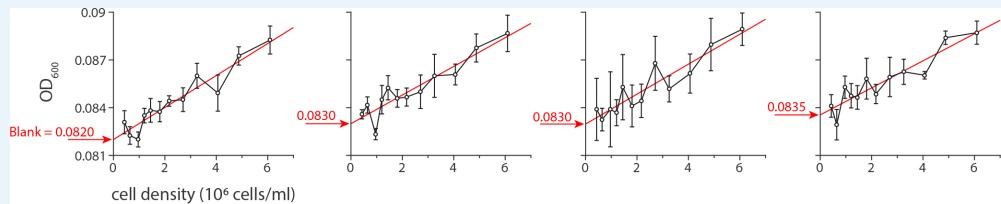

**Appendix 1—figure 2.** The value of the blank $OD_{600}$ was determined by extrapolating the relationship between the initial readings of the $OD_{600}$ generated by the plate reader against the known cell densities of the culture used in the microplate.

DOI: https://doi.org/10.7554/eLife.38174.014

## Appendix 2

DOI: https://doi.org/10.7554/eLife.38174.012

### Testing stability of antimicrobial peptide LL37 in 37°C temperature

The growth of some of the cultures in the late stages of an experiment raises the question that LL37 peptide may not remain stable throughout the duration of the experiment. To rule out this possibility, we conducted experiments to measure the MIC of LL37 peptide, incubated at 37°C for varying amounts of time including 0, 6, 12, 18 hr prior to the MIC experiment. The results do not show any dependence of MIC on the incubation time.

A linear range of drug concentrations from 0 to 2 $\mu$M was used for all trials. **Appendix 2—figure 1** depicts the microplate data and MIC measurements where, despite small fluctuations, the MIC for all of the incubation times was found to be in the range of 0.8–1.0 $\mu$M.

The microplate scheme includes one 'blank' column consisting of eight wells containing 100 $\mu$l of growth media (RMD) to ensure the absence of contaminants in the growth media. For columns 2 through 12, a linear concentration gradient of LL37 peptide was tested, ranging from 0.0 to 2.0 $\mu$M. Eight rows of the microplate were utilized to accommodate four different pre-incubation times of LL37 solution in 37°C as demonstrated in **Appendix 2—figure 1**. The cells were harvested from an early exponential culture at $OD_{600}$ = 0.2 and then diluted 100,000-fold. The red curve depicts growth, in terms of $OD_{600}$ over 24 hr. The grey curve corresponds to the wells where growth was inhibited, while the green border refers to the minimum concentration of LL37 peptide responsible for inhibiting growth.

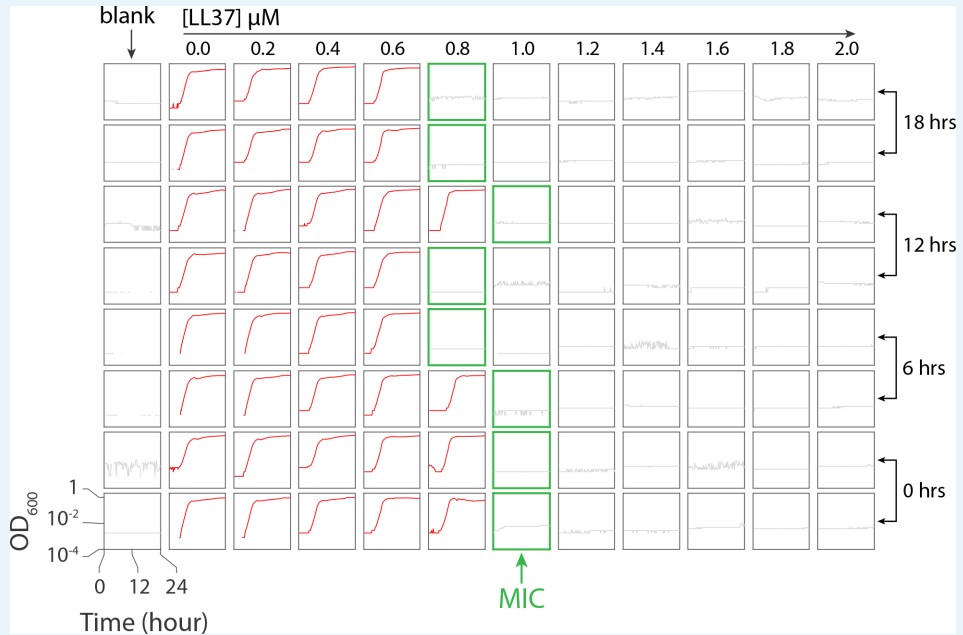

**Appendix 2—figure 1.** The microplate scheme for testing stability of LL37 in 37°C. The 'blank' column contains growth media to ensure the absence of contaminants in the media. Columns 2 through 12 contain a linear concentration gradient of LL37 from 0 to 2 $\mu$M. Four different pre-incubation times of LL37 solution in 37°C were tested. The red curve depicts bacterial growth, whereas the grey curve corresponds to wells for which growth was inhibited. Green borders refer to the MIC consisting of the lowest LL37 concentration responsible for inhibiting growth.

DOI: https://doi.org/10.7554/eLife.38174.016

## Appendix 3

DOI: https://doi.org/10.7554/eLife.38174.012

### Dye-tagged antimicrobial peptide sequestration experiment

To test the absorption capability of LL37 peptides by cells, we have conducted colorimetric experiments where *E. coli* cells were treated with a dye-tagged version of LL37 (5-FAM-LC-LL37). The concentration of the dye-tagged peptides was measured by fluorescence detection using a microplate reader (H1M synergy, BioTek) before and after treating cells. Specifically, cells were first treated with an above MIC concentration of dye-tagged peptides ($14\mu$M) and then removed from the culture via centrifugation. The density of the dye-tagged peptides was measured afterwards. The reduction in the peptides' density in the solution revealed the amount of peptides that were absorbed by the cells. While testing various cell densities and keeping the initial concentration of the peptides constant, we were able to establish a relationship between the inoculum size and the amount of absorbed peptides by the cells.

A total of three different cell densities were tested as well as a control lacking cells to prove that no peptides were lost throughout the experiment. The cells were harvested in mid-exponential phase and the measured densities included 1.5, 0.76, and $0.38 \times 10^6$ cells/ml.

The experiment was performed in four steps. First, the solution of 5-FAM-LC-LL37 in the growth media (RDM) and the cell culture were transferred to the microplate wells as depicted in *Appendix 3—figure 1*. Three replicates of the wells were performed and are shown in the figure as well. Second, the plate was incubated at 37°C while shaking at 590 rpm. Next, the culture was collected in microcentrifuge tubes and centrifuged at 1,000 rpm for one minute. The supernatant was then collected and transferred to another plate for reading fluorescent signals. Lastly, the fluorescent signal was used to infer the peptide density after calibration of the fluorescence intensity with standards with varying, known peptide concentrations.

While the initial concentration of the peptides was selected to be 14 $\mu$M, the final concentrations of each sample seemed to be varying, which was dependent on the cell density in each well as depicted in *Figure 1F* of the main text.

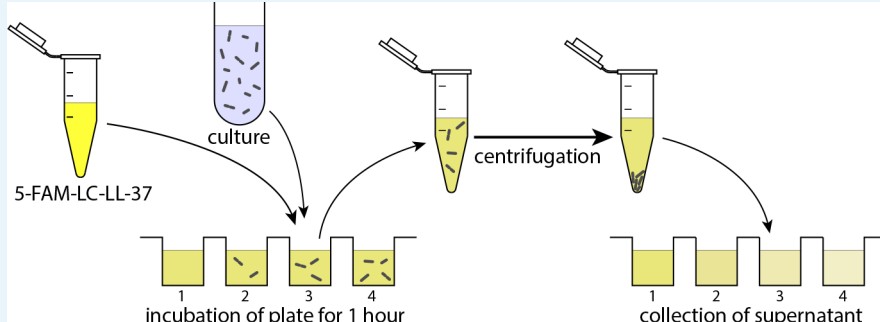

**Appendix 3—figure 1.** The schematic illustration of the experimental procedure for measuring the remaining peptides in the solution after absorption by bacteria. First, the solution of 5-FAM-LC-LL37 in RDM and the cell culture is transferred to and incubated at 37°C in a microplate. The culture is then collected and centrifuged. The supernatant is transferred to another plate for fluorescence reading. The fluorescent signal is correlated with the peptide density. The final concentration of peptides (wells on the right) is dependent on the cell density.

DOI: https://doi.org/10.7554/eLife.38174.018

