## [Decision Letter]

Thank you for submitting your article "Heterogeneous Absorption of Antimicrobial Peptide LL37 in *Escherichia coli* Cells Enhances Population Survivability" for consideration by *eLife*. Your article has been reviewed by three peer reviewers, and the evaluation has been overseen by Ry Young as a Guest Reviewing Editor and Gisela Storz as the Senior Editor. The following individual involved in review of your submission has agreed to reveal his identity: William Wimley (Reviewer #2).

The reviewers have discussed the reviews with one another and the Reviewing Editor has drafted this decision to help you prepare a revised submission.

Summary:

Snoussi et al. have presented a straightforward study of the "inoculum effect" for the classical human AMP LL37. The authors test the hypothesis denoted in the title using LL37 against the Gram-negative bacterial species. *E. coli*. The authors report that the peptide is absorbed by part of the cells in the population, allowing the remaining cells to grow, which results in an increase in MIC with the innoculum size. The authors study and model the binding and accumulation of LL37 on the growth and viability of *E. coli* cells. Importantly, they do this quantitatively at the level of single cells and ensembles. They directly measure the binding of LL37 to *E. coli*, providing an important measurement that is usually lacking. Authors study the inoculum effect and successfully model the effect of peptide concentration and inoculum size. This paper presents several critical observations that will be of great interest to AMP researchers. Perhaps most importantly, the authors approach this problem in a highly quantitative manner, providing models that may enable the prediction of AMP behavior. This is rare and it may, in turn, assist in future engineering and development of novel AMPs. The quality of the data are excellent and the conclusions interesting, surprising, and potentially of general applicability to understanding of the antimicrobial effects of AMPs and other compounds. This is a very strong paper that provides deep, novel insights into a well-studied field that usually does not provide new insights. Thus, this is an important paper, and should be published.

Essential revisions:

1) There may be two opportunities for the authors to buttress their case. First, in the experiment shown in Figure 1, shouldn't it be possible to estimate the number of cells that are susceptible under each condition and correlate that with the observed delay times? This could be done in the Discussion, after the apparent permeability collapse has been documented. Another step that should be doable is to force the cell populations into non-septating cell forms, either by judicious use of an antibiotic, the induction of a phage septation inhibitor like lambda Kil, or by using ts septation mutants, shifted to the non-permissive temperature. We would then expect for there not to be two populations of fluorescently labelled cells and the inoculum effect would presumably be abolished.

2) Figure 1 also should be improved. Figure 1E suggests the supernatant concentration of the dye-tagged LL37 is being measured. The text claims absorption of LL37 into the cell so the colorimetry readout of the supernatant fraction should decrease with increasing inoculum size. However, the readout in 1F is absorbed LL37. Is this inferred from subtracting the final signal of LL37 from the initial? The calculation should be defined explicitly and fluorescence measurements of both the supernatant and pellet should be provided. Theoretically fluorescence of the pellet + supernatant should add to the initial fluorescence unless there is degradation.

3) The authors have another opportunity to correlate binding with targets in the cell. In Figure 6, the absorption kinetics of LL37 into enucleated minicells budding from mother cells was qualitatively compared to cells with DNA. A graph of adsorption into the minicell was presented in 6B. Although retention appears unaffected, it appears the binding rate is slower in the enucleated cell. If this is so, it should be addressed in the text, since the authors are trying to evaluate what the target molecules in the cytoplasm and the presence and absence of DNA. Figure 6B could also include a comparative trace of adsorption kinetics into the neighboring nucleated cells, as perfect controls. If the rate was slower for smaller cells it could support conclusions from 5A.

4) Please address the following point in the manuscript text: A major problem with this study is the very modest effect of the inoculum size on the MIC, a 1.5 fold increase (Figure 1B). The accepted variation in the determination of MIC is 2 fold. Of course, it is possible to perform a more accurate measurement of MIC with many replicates and using small increments of increase in antimicrobial, which the authors do in this study. However, the significance of this small effect on the biology or clinical manifestations of an infection is unclear.

---

## [Author Response]

Essential revisions:1) There may be two opportunities for the authors to buttress their case. First, in the experiment shown in Figure 1, shouldn't it be possible to estimate the number of cells that are susceptible under each condition and correlate that with the observed delay times? This could be done in the Discussion, after the apparent permeability collapse has been documented.

The observed time delay in growth of cells is related to the number of non-growing cells, which is a function of the initial concentration of AMPs. It is important to note that our experiments show that sometimes cells divide several times before they stop growing. This makes it hard to directly relate the number of non-growing cells to the observed time delay. Our mathematical model captures the possibility that cells stop growing at different generations and was used to compare the predicted growth delay with experimental measurements (see Figure 4D). The predictions of our model agree reasonably well with experimental results. Deviations are likely coming from the fact that cell doubling times are scattered (Figure 1D), while our model assumes a fixed cell doubling time of 23 minutes. Due to the exponential growth, the growth delay is highly sensitive to the cell doubling time. This is stated explicitly in text (final paragraph subsection “Mathematical model based on peptide absorption reproduces and explains experimental observations”).

Another step that should be doable is to force the cell populations into non-septating cell forms, either by judicious use of an antibiotic, the induction of a phage septation inhibitor like lambda Kil, or by using ts septation mutants, shifted to the non-permissive temperature. We would then expect for there not to be two populations of fluorescently labelled cells and the inoculum effect would presumably be abolished.

This is a stimulating comment that led us design a new experiment for the manuscript. We added a new figure (Figure 5B) and revised the text (last paragraph of subsection “The action of dye-tagged LL37 peptides is cell-cycle and cell size dependent”) in connection with the new data. The comment correctly points out that non-septating cells should have higher resistance to AMPs. The difficulty with non-septating cells is that they are not viable as they don’t divide in daughter cells. Thus, it is hard to measure any MIC value corresponding to the non-septating cells.

Instead, we have forced cells to delay division via administering a sub-lethal dosage of cephalexin (1µg/ml). With this dosage cells growing in rich growth media (RDM) become longer in length without any appreciable change in the growth rate or cell doubling times. This has been reported by Fangwei Si, et al. (Current Biology 2017) and we have also confirmed it in our lab with the microfluidics mother machine.

We have measured the MIC using a 96-well plate with the inoculum size of 6.09×10^6^ cells/ ml for cells growing in RDM and those growing in RDM + 1 µg/ml cephalexin. Conducting three biological repeats with a total of 12 replicates for each condition we found that the delayed division of the cells via cephalexin increases the MIC from 1.87 ± 0.23µM to 2.09 ± 0.19µM (Figure 5B).

2) Figure 1 also should be improved. Figure 1E suggests the supernatant concentration of the dye-tagged LL37 is being measured. The text claims absorption of LL37 into the cell so the colorimetry readout of the supernatant fraction should decrease with increasing inoculum size. However, the readout in 1F is absorbed LL37. Is this inferred from subtracting the final signal of LL37 from the initial? The calculation should be defined explicitly and fluorescence measurements of both the supernatant and pellet should be provided. Theoretically fluorescence of the pellet + supernatant should add to the initial fluorescence unless there is degradation.

Reviewers are correct. The amount of absorbed AMPs is inferred by subtracting the final concentration of AMPs from the initial value. We have revised the Figure 1F to include both the remaining concentration in the supernatant and the inferred absorbed concentration. The caption was also revised accordingly:

“The amount of absorbed AMPs by the cells are inferred by subtracting the final

(supernatant) from the initial concentration of AMPs.”

and discussed it in the text:

“We observed a reduction in supernatant concentration of AMPs proportional to the inoculum size with an average rate of 7.6 ± 2.1 ×10^8^ AMPs/cell (calculated based on a linear fit to the data). Since the colorimetric measurements rely on clear solutions, we had to infer the absorbed AMPs by subtracting the supernatant concentration from the initial concentration (Figure 1F left axis).”

This approach was used, because cells (whether live or dead) are opaque objects and their presence affects the reading of the fluorescence signal by blocking and diffracting the light. Thus, the calibration of the method and measurements were done with clear solutions.

Because of this, we are limited to measuring the concentration of AMPs in the supernatant.

3) The authors have another opportunity to correlate binding with targets in the cell. In Figure 6, the absorption kinetics of LL37 into enucleated minicells budding from mother cells was qualitatively compared to cells with DNA. A graph of adsorption into the minicell was presented in 6B. Although retention appears unaffected, it appears the binding rate is slower in the enucleated cell. If this is so, it should be addressed in the text, since the authors are trying to evaluate what the target molecules in the cytoplasm and the presence and absence of DNA. Figure 6B could also include a comparative trace of adsorption kinetics into the neighboring nucleated cells, as perfect controls. If the rate was slower for smaller cells it could support conclusions from 5A.

The comment correctly points out that the rates of adsorption of AMPs to the nucleated and enucleated cells are different, even though the retention is qualitatively similar. For a direct comparison we have added the fluorescence data for the nucleated neighboring mother cells and also added data for another mini-cell in the revised Figure 6B. In the revised version of the text we specifically mentioned that:

“Translocation of 5-FAM-LC-LL37 peptides into mini-cells showed qualitatively similar absorption and retention as seen in regular cells (Figure 6B), which suggests the presence of significant interactions of AMPs with the intracellular content other than DNA. Yet, the rate at which AMPs are absorbed into the mini-cells is slower than that in the neighboring mother cells with DNA content (Figure 6B), perhaps indicating the role of the negative charge of DNA.”

and in the caption of Figure 6 we mentioned:

“Fluorescence intensity per area of two mini-cells suggests a significant peptide interactions with enucleated mini-cells. The rate of absorption of AMPs in mini-cells is slower than in the neighboring mother cells with DNA content.”

We believe that identifying the exact binding target of AMPs is a complicated task that is beyond the scope of this work. The reason behind the difference of binding rates between the nucleated and enucleated could be more than just the DNA content. The ribosome content and proteome may also play an important role, which is yet to be determined.

4) Please address the following point in the manuscript text: A major problem with this study is the very modest effect of the inoculum size on the MIC, a 1.5 fold increase (Figure 1B). The accepted variation in the determination of MIC is 2 fold. Of course, it is possible to perform a more accurate measurement of MIC with many replicates and using small increments of increase in antimicrobial, which the authors do in this study. However, the significance of this small effect on the biology or clinical manifestations of an infection is unclear.

The range of the MIC in Figure 1B was very narrow, only 1.5 fold, because we limited the assay to dilute cultures in order to avoid any side effects of stationary phase cultures (The highest cell density in the 96-well plate scheme of Figure 1A is 6.09×10^6^ cells/ml). The reason is that we wanted to gain insights about a single-cell behavior within a population. To demonstrate the importance of the inoculum effect beyond this modest range, we conducted additional MIC assays for higher inoculum size showing the MIC can increase by about one order of magnitude. In the revised version of the text we reported that:

“A separate set of experiments with dense cultures showed that the MIC increases to (𝟥.𝟨𝟫 ± 𝟢.𝟦𝟥) μ𝖬 and (𝟩.𝟢𝟫 ± 𝟣.𝟪𝟪) μ𝖬 for the inoculum sizes of 𝟣𝟤.𝟤×𝟣𝟢𝟨 and 𝟤𝟦.𝟦×𝟣𝟢𝟨 cells/ml (8 replicates with 3 biological repeats were used for each reported value). This is one order of magnitude increase in the MIC which can be critical in medical applications.”